# Nitric Oxide Participates in Aluminum-Stress-Induced Pollen Tube Growth Inhibition in Tea (*Camellia*
*sinensis*) by Regulating *CsALMTs*

**DOI:** 10.3390/plants11172233

**Published:** 2022-08-29

**Authors:** Xiaohan Xu, Zhiqiang Tian, Anqi Xing, Zichen Wu, Xuyan Li, Lingcong Dai, Yiyang Yang, Juan Yin, Yuhua Wang

**Affiliations:** 1College of Horticulture, Nanjing Agricultural University, Nanjing 210095, China; 2Institute of Leisure Agriculture, Jiangsu Academy of Agricultural Sciences, Nanjing 210014, China; 3Jiangsu Maoshan Tea Resorts Company Limited, Changzhou 213200, China

**Keywords:** tea plant, pollen tube, nitric oxide, aluminum stress, aluminum-activated malate transporter

## Abstract

Nitric oxide (NO), as a signal molecule, is involved in the mediation of heavy-metal-stress-induced physiological responses in plants. In this study, we investigated the effect of NO on *Camellia sinensis* pollen tubes exposed to aluminum (Al) stress. Exogenous application of the NO donor decreased the pollen germination rate and pollen tube length and increased the malondialdehyde (MDA) content and antioxidant enzyme activities under Al stress. Simultaneously, the NO donor effectively increased NO content in pollen tube of *C. sinensis* under Al stress and could aggravate the damage of Al^3+^ to *C. sinensis* pollen tubes by promoting the uptake of Al^3+^. In addition, application of the NO-specific scavenger significantly alleviated stress damage in *C. sinensis* pollen tube under Al stress. Moreover, 18 *CsALMT* members from a key Al-transporting gene family were identified, which could be divided into four subclasses. Pearson correlation analysis showed the expression level of *CsALMT8* showed significant positive correlation with the Al^3+^ concentration gradient and NO levels, but a significant negative correlation with pollen germination rate and pollen tube length. The expression level of *CsALMT5* was negatively correlated with the Al^3+^ concentration gradient and NO level, and positively correlated with pollen germination rate and pollen tube length. The expression level of *CsALMT17* showed a significant negative correlation with Al^3+^ concentration and NO content in pollen tubes, but significant positive correlation with pollen germination rate and pollen tube length. In conclusion, a complex signal network regulated by NO-mediated *CsALMTs* revealed that *CsALMT8* was regulated by environmental Al^3+^ and NO to assist Al^3+^ entry into pollen tubes; *CsALMT5* might be influenced by the Al^3+^ signal, stimulate malate efflux in vacuoles and chelate with Al^3+^ to detoxify Al in *C. sinensis* pollen tube.

## 1. Introduction

Aluminum (Al) is the most abundant metal in the Earth’s crust (comprising approximately 6.6% of the soil), and it is also an indispensable inorganic mineral in the soil [1]. Stable Al salt will become free Al ions (Al^3+^) that can be taken up by plants when the pH of the soil is lower than 5, which will affect plant growth [2]. As an acid-loving plant, *Camellia sinensis* (L.) O. Kuntze is better-suited to growing in acidic soil (pH 4.5–5.5), which has higher bioavailable Al in comparison to other plants. Thus, *C. sinensis* can absorb and accumulate more Al^3+^. In general, the root tip is the main part of plants responding to Al stress and generating a stress response [3]. On the one hand, aluminum stress can not only inhibit the absorption and metabolism of mineral ions such as Ca^2+^, Mg^2+^, K^+^, Fe^2+^, Cu^2+^, PO_4_^3-^ and water in plant root tip cells, but also inhibit the elongation and cell division of plant root tip cells [4]. It can also form Al-ATP complexes to inhibit the cytoplasmic membrane of plant roots by affecting the structure and function of calmodulin [5], thereby disrupting the Ca^2+^ balance and Ca^2+^ concentration gradient in plant cells. On the other hand, Al^3+^ can not only damage the plasma membrane of plants, but also change the functional structure of the cytoskeleton by affecting the morphological changes in microtubules and microfilaments [6]. Pollen tube, as the male reproductive organ of higher plants, plays an important role in sexual reproduction. The pollen tube grows from the top of style and presents a polarized pattern, which is a typical polar top growth, which is very similar to the growth of plant root tips [7]. In addition, the polar growth of pollen tubes is closely related to the cytoplasmic Ca^2+^ concentration gradient homeostasis and the morphology of microtubules and filaments [8].Therefore, pollen tubes provide a useful system for studying the mechanism of Al toxicity in plants. Many reports have shown that Al toxicity inhibits the growth of pollen tubes of tomato [9], lily [10], tea [11] and apple [12], but the mitigation measures for Al poisoning in pollen tubes have not been effectively explored.

Nitric oxide (NO) is involved in the regulation of plant growth and development, which is mainly reflected in seed development, root morphogenesis and stomatal movement [13,14]. Numerous studies in recent years have confirmed that NO is a crucial signal molecule for plants’ ability to deal with various challenges during growth [15,16]. In our previous studies, NO has been shown to play a role as a negative regulating factor in the response of *C. sinensis* pollen tubes to low-temperature stress [8]. In addition, some studies have demonstrated that applying an exogenous NO donor can significantly alleviate the Al stress on plants, such as alleviating the inhibition of Al stress on root length and reducing the accumulation of Al^3+^ in root tips [17]. NO has dual effects in relieving oxidative stress. On the one hand, NO reduces the accumulation of reactive oxygen species (ROS) by acting as an antioxidant and interacting with different ROS groups. On the other hand, NO can regulate the activity of antioxidant enzymes in plants and indirectly eliminate ROS accumulation induced by metal ions. Furthermore, NO can improve the capacity to alleviate Al stress by altering the polysaccharide components of rice cell walls [18].

Exploring the mechanism of plant Al resistance from genetic traits is an important way to screen Al-resistant genotypes. Many achievements have been made in elucidating the molecular mechanism of Al tolerance in plants, and the functions of aluminum-activated malate transporter (ALMT), a kind of Al-tolerance-related organic acid transporter, have been confirmed [19,20]. *TaALMT1* is structurally expressed in wheat root tips, and can be activated by Al^3+^ in acidic soil, releasing malate anion into the apoplast [21], and then utilizing malate to chelate Al^3+^, thereby improving the Al tolerance of wheat [20]. The release of organic acids (malate, oxalate, citrate, etc.) can relieve Al toxicity by chelating and immobilizing Al^3+^ has been confirmed in rice, barley, buckwheat, etc. In addition, members of the *ALMT* gene family also show similar Al tolerance in *Arabidopsis thaliana* [22], *Brassica napus* [23], *Glycine max* [24] and *Medicago sativa* [25]. To date, the *ALMT* gene family has been deeply studied in other plants, but rarely in *C. sinensis*. Pollen tube, as the male reproductive organ of higher plants, has a multinuclear single-cell structure, which is an ideal model to study the response of plants to various signals and stresses. In our pre-experiment, we found that *ALMTs* were also expressed in *C. sinensis* pollen tubes, and the expression levels of some *CsALMTs* in *C. sinensis* pollen tubes were higher than those in tea roots.

In this regard, we hypothesized that: (1) the Al tolerance of tea pollen tubes can be improved by increasing or decreasing NO levels in the environment; and (2) some *CsALMTs* can enhance the Al tolerance of *C. sinensis* pollen tubes through the mediation of NO signaling molecule tolerance. Thus, the present study aimed to explore the possible Al tolerance mechanism mediated by NO signals and *CsALMTs*, taking tea pollen tube as the research material for which the determination of physiological and biochemical indices, bioinformatics analysis and gene expression level analysis would be carried out.

## 2. Results

### 2.1. Pollen Germination and Pollen Tube Elongation

Exogenous Al^3+^ had a clear dose-dependent influence on the pollen germination rate and pollen tube length, both of which significantly decreased with an increase in the Al concentration gradient (*p* < 0.05) (Figure 1A,B and Appendix A). Moreover, the pollen germination rate and pollen tube length were considerably reduced by exogenous NO donor treatments compared to Al treatments, and the inhibitory effect of NO on pollen germination and growth increased with the increase in the Al concentration gradient (Figure 1A). Furthermore, Al + cPTIO treatments demonstrated a favorable impact on the pollen germination rate and pollen tube length when compared to Al treatments, and the alleviating effect of cPTIO on the exogenous Al^3+^-inhibited reduction in the pollen germination rate and pollen tube length of *C. sinensis* also increased with the increase in the Al concentration gradient (Figure 1A,B).

### 2.2. Cytoplasmic Al^3+^ and NO

The results show that cytoplasmic Al^3+^ and NO concentrations were significantly increased under the treatment of exogenous Al^3+^, and cytoplasmic Al^3+^ and NO concentrations reached the highest value under 2.5 mM Al^3+^ treatment (Figure 1C,D). Additionally, the pollen tube Al^3+^ and NO concentrations of Al1 + NO and Al2.5 + NO were significantly higher than those of Al1 and Al2.5, respectively, while the cytoplasmic Al^3+^ and NO concentrations of Al1 + cPTIO and Al2.5 + cPTIO were significantly lower than those of Al1 and Al2.5, respectively (Figure 1C,D).

### 2.3. Physiochemical Responses

Malondialdehyde (MDA) contents in pollen tubes increased with the increasing Al concentration gradient (Figure 2A). It is worth noting that an exogenous addition the NO donor and NO scavenger increased and decreased MDA contents in pollen tubes, respectively. However, compared with Al0, Al0 + NO increased the MDA contents in pollen tubes by 21.98% but Al0 + cPTIO decreased the content by 16.80%, while there was no significant difference in MDA contents among Al5, Al5 + NO and Al5 + cPTIO (*p* > 0.05).

The activities of antioxidant enzymes in pollen tubes were significantly influenced by the Al concentration gradient and NO level (Figure 2B–D). The activities of all three antioxidant enzymes (superoxide dismutase (SOD), catalase (CAT) and peroxidase (POD)) in Al, Al + NO and Al + cPTIO showed the same trend with the exogenous Al concentration gradient; that is, they increased until reaching the highest activity under the 2.5 mM Al^3+^ treatment and then decreased. In addition, the NO donor/NO scavenger could promote/inhibit the activities of SOD, CAT and POD in pollen tube only when the concentrations of Al^3+^ in the culture medium were 1 mM and 2.5 mM.

### 2.4. Bioinformatics Analysis for ALMT Gene Family in C. sinensis

#### 2.4.1. Identification, Classification and Naming of ALMT Gene Family in *C. sinensis*

By comparing the genomic coding sequences of *C. sinensis* and the *AtALMTs* sequences using Local BLASTp and Bioedit, 18 potential ALMT amino acid sequences in *C. sinensis* were discovered. The ALMT conserved domain (PF11744) of the obtained sequences was verified again by online services SMART and Pfam. It was found that all 18 amino acid sequences contained the ALMT conserved domain, and finally, 18 ALMT amino acid sequences in *C. sinensis* were obtained.

In the present study, the phylogenetic tree was constructed by using amino acid sequences of *CsALMTs*, *AtALMTs* and *OsALMTs* (Appendix A). The result of the phylogenetic tree showed that *ALMTs* could be divided into four subclasses, among which subclass I includes 5 *AtALMTs*, 4 *OsALMTs* and 10 *CsALMTs*. Subclass II includes five *AtALMTs*, four *OsALMTs* and four *CsALMTs*. Subclass III includes four *AtALMTs*, one *OsALMT* and two *CsALMTs*. It is worth noting that only *CsALMT17* and *CsALMT18* are included in subclass IV (Appendix A). Previous studies have shown that *AtALMT1* encodes Al-activated root malate efflux transporter and is related to Al tolerance [26]. Therefore, we speculate that *CsALMTs* in subclass I may be related to Al resistance.

#### 2.4.2. Basic Physicochemical Properties, Subcellular Localization and Protein Secondary Structure Prediction of CsALMTs

Based on fundamental physical and chemical features, it was determined that each of the 18 CsALMTs encoded a different number of amino acids, with *CsALMT18* encoding the fewest (302 amino acids) and *CsALMT5* encoding the most (793 amino acids) (Appendix A). The molecular weights of 18 *CsALMTs* ranged from 32,991.54 to 88,719.08 Da, and the theoretical isoelectric points ranged from 5.66 to 8.98. Total average hydrophobicity ranged from −0.152 to 0.421, and protein instability coefficient ranged from 28.24 to 41.93. The analysis of the number of transmembrane proteins showed that *CsALMTs* contained three to six transmembrane proteins except *CsALMT18*, which contained no transmembrane domain. The subcellular localization of members of this gene family was predicted by SoftBerry ProtComp 9.0, and it was found that *CsALMT1*, *CsALMT2* and *CsALMT6-16* were all located on the plasma membrane, *CsALMT3-5* was located on the endoplasmic reticulum, and *CsALMT17* and *CsALMT18* were located outside the cells.

The protein secondary structures of CsALMTs were predicted using the web program SOPMA (Appendix A). The results show that the protein secondary structure of all the 18 CsALMTs contained an α helix, β turn, irregular curl and extended chain. Among them, the α helix (36.47–64.11%) was the main structure in CsALMTs, followed by irregular coiled structure and an extended chain. The β-turn accounts for the lowest number of the secondary structures in CsALMTs, ranging from 1.88% to 11.06%. Irregular curly structure and extended chains accounted for 20.33–30.45% and 8.82–29.18%, respectively.

#### 2.4.3. Analysis of Gene Structure and Amino Acid Structure of CsALMTs

According to Appendix A, the majority of CsALMTs had six exons and were primarily found in subclasses I and II. The exon count for CsALMT5 was 11, whereas the exon count for CsALMT18 was only 4. Seventeen amino acid motifs of CsALMTs (Appendix A) were obtained using the MEME online service, and the E value and multilevel consensus sequence of each motif are shown in Appendix A. CsALMTs contains at least 2 motifs and at most 15 motifs, among which 12 motifs (motifs 1, 2, 3, 4, 5, 6, 7, 8, 9, 10, 11 and 14) were commonly included in subclasses I-III. Motif 1 was present in each CsALMT, and only the CsALMTs in subclasses I and II included motif 17 and motif 12, respectively. However, CsALMTs in subclass II did not contain motif 13, and CsALMTs in subclass III did not contain motif 15.

### 2.5. Expression Analysis of CsALMTs in Different C. sinensis Tissues

To explore the expression differences in *CsALMTs* in different tissues of *C. sinensis*, RT-qPCR was used to analyze the expression patterns of *CsALMTs* in four different tissues (root, stem, leaf, flower and pollen tube) of *C. sinensis*. The results show that the expression of 18 *CsALMTs* had tissue specificity (Figure 3). All 18 CsALMTs were expressed in the root and stem, but *CsALMT2*, *7*, *10*, *13* and *15* were not expressed in leaves, *CsALMT7* and *11* were not expressed in flowers, and *CsALMT3*, 6, *7*, *11*, *12*, *13*, *14* and *15* were not expressed in pollen tubes. Among the genes which had significantly higher expression levels in the stem than in the root (*CsALMT3*, *4*, *6*, *7*, *8*, *9*, *11*, *14*, *15* and *17*), *CsALMT3*, *6*, *9* and *17* and *CsALMT3*, *8*, *9*, *14*, *15* and *17* also had significantly higher expression levels in leaves and flowers than in the root, respectively. In addition, only CsALMT18 had a significantly higher expression level in pollen tubes than in the root.

### 2.6. Effect of Al^3+^ and NO on CsALMTs’ Expression in Pollen Tube

In order to explore the expression level of *CsALMTs* in *C. sinensis* pollen tubes under different treatments, RT-qPCR technology was also used to quantitatively analyze *C. sinensis* pollen tubes.

In Al treatment, Al + NO and Al + cPTIO, the expression level of *CsALMT1* and *CsALMT8* increased with the increased Al concentration gradient (Figure 4). Compared with Al treatment, cPTIO significantly increased the relative expression level of *CsALMT1* (*p* < 0.05), but had no significant effect on *CsALMT8*; however, the relative expression levels of *CsALMT8* in Al0 + NO and Al1 + NO were significantly higher than those in Al0 and Al1, respectively. In Al treatment and Al + NO, the concentration gradient of Al^3+^ had no significant effect on the expression level of *CsALMT10*. However, in Al + cPTIO, the expression level of *CsALMT10* increased until the highest expression level under Al2.5 + cPTIO was reached and then decreased. Furthermore, both the NO donor and NO scavenger enhanced the expression level of *CsALMT16* with 0 mM and 1 mM Al^3+^ in the culture medium.

The expression levels of *CsALMT4*, *5* and *9* in subclass II did not show an obvious trend along the Al concentration gradient (Figure 4). However, it should be noted that the expression levels of *CsALMT4* and *CsALMT9* in Al0 + NO and Al0 + cPTIO were significantly lower than those of Al0. Similar to *CsALMT4* and *CsALMT9*, the expression levels of *CsALMT17* and *CsALMT18* were significantly lower than those of the control under normal cytoplasmic Al^3+^ concentration after the addition of the NO donor and NO scavenger. Moreover, the expression levels of *CsALMT17* in Al + cPTIO were significantly higher than those of the other two treatment groups after the addition of Al^3+^. However, for *CsALMT18*, only the expression level in Al1 + NO was significantly higher than that in Al1 and Al1 + cPTIO.

### 2.7. Two-Way ANOVA Analysis and Pearson Correlation Coefficients

In order to further clarify the effects of the Al concentration gradient and NO level on physiological and biochemical indices and the *CsALMTs* expression level in *C. sinensis* pollen tube, two-way ANOVA was performed (Table 1). One the one hand, the Al concentration gradients and NO levels were highly significantly (*p* < 0.001), and influenced all eight physiological and biochemical indices. Aside from the cytoplasmic NO concentration, the other seven indices were affected by the Al concentration gradient to a higher degree than the NO level. Furthermore, the combined effects of the Al concentration gradient and NO level on the pollen germination rate and POD activity were not significant. On the other hand, the relative expression levels of *CsALMT1*, *2*, *4*, *5*, *8*, *9*, *10*, *16*, *17* and *18* were all significantly (*p* < 0.05) affected by Al concentration gradient and NO level. Among them, *CsALMT1*, *2*, *4*, *5*, *8*, *9* and *16* were more influenced by the Al concentration gradient than NO level, while *CsALMT10*, *17* and *18* were more influenced by the NO level.

The Pearson correlation analysis was carried out to further explore the role of *CsALMTs* in NO involved in the process of the *C. sinensis* pollen tube response to Al stress (Figure 5). The data for the 5 mM Al^3+^ treatment were eliminated due to the NO level, which had only a weak effect on the indicators of tea pollen tubes under the 5 mM Al^3+^ treatment. Three levels of NO were assigned: Al + cPTIO treatment = 0, Al treatment = 1 and Al + NO = 2. The expression levels of *CsALMT1*, *CsALMT8* and *CsALMT10* in subclass I showed a significant positive correlation with the Al concentration gradient. Among these three *CsALMTs* in subclass I, *CsALMT8* also showed a significant positive correlation with the NO level and cytoplasmic concentration of Al^3+^ and NO, but a significant negative correlation with the pollen germination rate and pollen tube length. *CsALMT1*, *10* and *17* showed a significant negative correlation with the NO level, and *CsALMT5* and *CsALMT17* showed a significant negative correlation with cytoplasmic Al^3+^ concentration (Figure 5).

## 3. Discussion

Depending on the timing and level of NO production in response to stress, plants may create NO endogenously and use it to respond in particular ways [27,28]. When NO concentrations rise under stressful circumstances, it is thought that it can potentially enhance cytotoxicity in addition to acting as a signal molecule [29,30]. In the present study, exogenous Al^3+^ significantly inhibited the pollen germination and the pollen tube elongation of *C. sinensis* in a dose-dependent manner (Figure 1). This result is consistent with the results reported for Al-stressed pollen tubes of apple [12]. In addition, the exogenous NO donor could inhibit pollen tube growth and enhance the inhibition of Al^3+^ for the pollen germination rate and pollen tube length, while exogenous cPTIO alleviated this inhibition. On the other hand, exogenous NO donors exacerbated the oxidative damage to Al3+ in pollen tubes, while exogenous NO scavengers lessened the oxidative damage to Al3+ in pollen tubes, as shown by MDA content (Figure 2A). These results, similar to those reported for alfalfa [31], wheat [32] and rice [33], suggest that the high NO level conferred Al sensitivity on pollen tube polar growth in *C. sinensis*. Moreover, there was no significant difference in cytoplasmic Al^3+^ concentration, SOD, CAT and POD activities among Al0, Al0 + NO and Al0 + cPTIO (Figure 1 and Figure 2), indicating that the NO level had a weak effect on the absorption of Al^3+^ by pollen tubes without exogenous Al addition. Furthermore, the encouragement of DEA NONOate and the inhibition of cPTIO on pollen tube absorption of Al^3+^ steadily increased with the increase in the Al concentration gradient. This suggests that the NO level in the plant growth environment could regulate the absorption of Al^3+^ by plants, and had a positive correlation, which further indicated that NO could participate in the response of *C. sinensis* pollen tube extension to Al stress as a negative regulator. Previous research had demonstrated that NO generation and an increase in NOS activity occurred in response to biotic or abiotic stress in plants. For example, according to Wang et al. [34], ultrasonography can cause Taxus cells to begin accumulating H_2_O_2_ and triggering programmed death, which can be facilitated by sodium nitroprusside (an NO donor), but partially blocked by an NO scavenger (cPTIO) and NOS scavenger (L-NNA). The significant positive correlation between NO concentration and cytoplasmic Al^3+^ concentration in pollen tubes indicates that the *C. sinensis* pollen tube is accompanied by NO production under Al stress. On the other hand, the activities of SOD, POD and CAT in pollen tube were significantly increased by the exogenous NO donor, while the activities of these antioxidant enzymes were decreased when cPTIO was added; however, when the Al concentration gradient was increased up to 5 mM, the cytoplasmic Al^3+^ concentration, SOD, CAT and POD activities decreased significantly in the comparison to when the gradient was 2.5 mM, and there was no significant difference among Al5, Al5 + NO or Al5 + cPTIO (Figure 1 and Figure 2). This seems to imply that (i) excessive NO could activate the antioxidant system of C. sinensis pollen tube, but that it could also cause oxidative stress; (ii) high concentrations of exogenous Al^3+^ severely inhibit the growth and development of C. sinensis pollen tube, even destroying the antioxidant system, and mask the influence of NO level.

Gene expression analysis is a very important part of gene function research. The expression analysis of the *ALMT* family in different tissues of rice has been studied. *OsALMT1*, *OsALMT7* and *OsALMT9*, which belong to subclass I, are expressed in the root of rice [21]. In this study, *CsALMT1*, *CsALMT2*, *CsALMT8*, *CsALMT10* and *CsALMT16*, which belonged to subclass I, were expressed both in the root and pollen tube (Figure 3), and the expression levels of all except *CsALMT2* showed an up-regulated pattern along the Al concentration gradient in Al treatment (Figure 4A). This was consistent with the expression of *HbALMT1*, *HbALMT2* and *HbALMT15*, which were involved in Al detoxification [35], suggesting that the *CsALMTs* in subclass I might play a role in alleviating Al stress. However, results of Pearson correlation analysis show that *CsALMT2* and *CsALMT10* only had a significant correlation with the NO level (Figure 5); therefore, we further ruled out the Al detoxification effect of *CsALMT2* and *CsALMT10* in the *C. sinensis* pollen tube. In addition, the expression levels of *CsALMT1* and *CsALMT8* under Al5 + cPTIO were significantly higher than those under other Al + cPTIO treatments, so we speculate that *CsALMT1* and *CsALMT8* might have synergistic effects under the induction of exogenous cPTIO to jointly alleviate Al stress. In addition, *CsALMT8* also showed a significant positive correlation with the NO level, cytoplasmic Al^3+^ concentration and cytoplasmic NO concentration, but a significant negative correlation with the pollen germination rate and pollen tube length, suggesting that *CsALMT8* participated in the response of *C. sinensis* pollen tube to Al stress as a negative regulator.

On the other hand, except for *CsALMTs* in subclass II, the expression levels of *CsALMT5* in subclass II and *CsALMT17* in subclass IV had a significant positive correlation with the pollen germination rate and pollen tube length (Figure 5). *CsALMT5* was also positively correlated with the concentration gradient of Al and the concentration of Al^3+^ in the cytoplasm. In the research of *AtALMTs* in subclass II, *AtALMT6* is aimed at the vacuole membrane of guard cells, which not only serves as the inflow and outflow channel of malate, but also controls the gas exchange; *AtALMT9* is a vacuole chlorine channel activated by malate, which plays an important role in the response of plant cells to the stomatal opening signaling pathway and contributes to the transportation of malate through the vacuole membrane [26]. Thus, we speculate that *CsALMT5* might alleviate the Al stress of the *C. sinensis* pollen tube by mediating the efflux of malate and then chelating or fixing extracellular Al^3+^. As for *CsALMT17*, it is not phylogenetically close to Al tolerance ALMT genes with a low response to the Al concentration gradient (Appendix A and Figure 5). Unexpectedly, *CsALMT17* showed a significant negative correlation with cytoplasmic Al^3+^ concentration. Since *CsALMT17* is subcellularly localized outside the cell, we speculate that it may be a secreted protein, which is synthesized by ribosomes on the surface of the endoplasmic reticulum and transported outside the cell. We speculate that Al^3+^ entering pollen tube cells could slow down the synthesis of CsALMT17 protein by inhibiting the expression of *CsALMT17*. In addition, the expression level of *CsALMT17* was increased by exogenous cPTIO and negatively correlated with the NO level and NO concentration in the cytoplasm, indicating that *CsALMT17* was negatively regulated by the cytoplasmic concentration of both Al^3+^ and NO. However, due to the lack of research on ALMT with extracellular subcellular localization, the relationship between this negative regulatory relationship and the promotion effect of *CsALMT17* on pollen germination and pollen tube growth are still unclear. Therefore, the mechanism of NO-mediated *CsALMT17* alleviating *C. sinensis* pollen tube Al toxicity still needs further study.

According to the results of this study, based on the comprehensive study of Al toxicity and tolerance in plants, the fundamental signal regulation network mainly relies on the NO and *CsALMTs* regulated, and their crosstalk is shown in Figure 6. On the one hand, *CsALMT8* localized on the plasma membrane can sense Al^3+^ outside pollen tubes and promote the uptake of Al^3+^, and this process is positively regulated by NO signaling molecules. After Al^3+^ enters the pollen tube, it accelerates the production of ROS, such as superoxide anions (·O_2_^-^), and then leads to programmed cell death through lipid peroxidation. These ROS are gradually converted into H_2_O_2_ and H_2_O under the action of SOD, POD and CAT. On the other hand, *CsALMT5* is negatively regulated both by the extracellular Al concentration gradient signal and cytoplasmic Al^3+^ concentration, which may hinder malate efflux from vacuoles, further weaken the chelation and fixation of Al^3+^ by extracellular malate and finally inhibit the growth of *C. sinensis* pollen tube. *CsALMT17* encodes an extracellular protein that is synthesized and secreted by cells and its expression level was negatively regulated by NO signaling molecules and cytoplasmic Al^3+^, and can promote the growth of *C. sinensis* pollen tube. In addition, Al^3+^ readily reacts with the negatively charged plasma membrane and ultimately disrupts the uptake of mineral ions, such as Ca^2+^ [36]. Additionally, it then affects the Ca^2+^ concentration gradient, which acts as an important factor regulating the polar growth of *C. sinensis* pollen tubes.

In summary, reducing or eliminating NO in the environment can be an effective way to enhance the Al resistance of *C. sinensis*; however, the regulatory effect of NO seems to be extremely weak in the case of a high Al^3+^ concentration in the environment. In addition, *CsALMTs* also played different roles under Al stress: *CsALMT8* was regulated by environmental Al^3+^ and NO to assist Al^3+^ entry into pollen tubes, while the up-regulated expression of *CsALMT5* could prevent Al^3+^ from entering pollen tubes by promoting malate efflux. This study also provides a new idea for using exogenous regulation of the NO level as a means to reduce Al content in tea.

## 4. Materials and Methods

### 4.1. Pollen Source and Culture

The mature pollen from 5-year-old *C. sinensis* cv. ‘Longjing43′ was collected on the day before blooms in November 2020. The *C. sinensis* plantation was located at Sun Yat-sen Tea Factory in Jiangsu Province, China (32°2′57′’ N, 118°50′35′’ E). The detailed information of in vitro pollen culture condition was mainly based on the previous publication with slight modifications [8]. Briefly, the pre-incubation of pollen was abandoned and directly treated in the dark at 25 °C for 60 min. For normal NO level treatments, the pollen was only treated with four Al concentration gradients (0 mM, 1 mM, 2.5 mM and 5 mM Al_2_(SO_4_)_3_·18H_2_O), and was marked as Al0, Al1, Al2.5 and Al5, respectively. For high-NO-level treatments, the pollen was treated with 25 µM diethylamine nonoate (DEA NONOate, an NO donor) and the above four Al concentration gradientsand was marked as Al + NO (Al0 + NO, Al1 + NO, Al2.5 + NO and Al5 + NO, respectively. For low-NO-level treatments, the pollen was treated with 200 µM 2-(4-carboxyphenyl)-4,4,5,5-tetramethylimidazoline-1-oxyl-3-oxide (cPTIO, an NO scavenger) and the above four Al concentration gradients, and was marked as Al + cPTIO (Al0 + cPTIO, Al1 + cPTIO, Al2.5 + cPTIO and Al5 + cPTIO, respectively. Three biological replicates were performed for each treatment. A portion of the culture solution containing pollen tubes was sucked after the pollen incubation process was complete in order to measure the pollen germination rate and pollen tube length. The remaining pollen tubes were filtered to remove ungerminated pollen grains and culture material using a nylon sieve with a hole size of 0.74 m, frozen in liquid nitrogen and then kept at −80 °C.

### 4.2. Observation of Pollen Germination Rate and Pollen Tube Elongation

To measure the mean pollen germination rate and pollen tube length, approximately 50 pollen tubes were detected in each of the three replicates after different treatments using a fluorescence microscope (DM6B, Leica, Wetzlar, Germany) and Image J (version 1.8.0, National Institutes of Health, Bethesda, MD, USA).

### 4.3. Measurement of Cytoplasmic Al^3+^ and NO

The cytoplasmic Al^3+^ concentration in the pollen tubes was measured as described by Havlin and Soltanpour with slight modifications [37]. Briefly, oven-dried and ground 100 mg pollen tube samples were digested with 5 mL of HNO_3_ and analyzed using inductively coupled plasma optical emission spectrometry (ICP-OES, iCAP 7400, Thermo Fisher, Waltham, MA, USA). The cytoplasmic NO concentration in pollen tubes were determined according to the operating instructions provided in the NO determination kit (Beyotime Biotechnology, Shanghai, China, Cat No. S0023).

### 4.4. Measurement of Cytoplasmic MDA and Antioxidant Enzyme Activity

The cytoplasmic MDA in pollen tubes was analyzed spectrophotometrically as described by Lei et al. [38] with slight modifications. In brief, 0.100 g of fresh sample was homogenized in 1 mL of 10% trichloracetic acid (TCA). Five hundred microliters of the supernatant was mixed with 500 μL of 10% TCA containing 0.5% thiobarbituric acid (TBA), and the mixture was heated in boiling water for 15 min. After rapid cooling, it was then centrifuged at 4 °C for 10 min at 10,000 rpm. The absorbance values were determined spectrophotometrically at 450, 532 and 600 nm by a multidetection microplate reader (CYTATION3, BioTek, Winooski, VT, USA).

For the determination of antioxidant enzyme activities in pollen tubes, 0.1 g of fresh sample was ground with 2 mL of 0.05 mM phosphate buffer (pH 7.8) in the ice bath. After grinding, the homogenate was centrifuged at 4 °C for 30 min at 10,000 rpm, and the supernatant was kept at 4 °C to be tested for enzyme activity.

Superoxide dismutase (SOD) activity was determined by measuring the inhibition of the photochemical reduction on pyrogallol utilizing the pyrogallol autoxidation method by spectrophotometry at 325 nm [39]. In brief, two test tubes were taken; one was added with 2.98 mL 50 mM Tris-HCl and 0.02 mL 10 mM HCl, and the other contained 2.98 mL 50 mM Tris-HCl, 0.2 mL 10 mM HCl and 50 mM pyrogallol mixture. The absorbance changes (ΔOD) were determined within 4 min by spectrophotometry at 325 nm. The volume of pyrogallol was fixed in the mixture to obtain an OD value at 0.07. In the determination of enzyme activity, 0.01 mL of HCl was replaced with 0.01 mL of enzyme solution, and ΔOD was determined spectrophotometrically at 325 nm for 4 min and recorded every 30 s. The SOD activity was calculated using the following formula:$$\mathrm{SOD}\;\mathrm{activity}\;=\frac{0.07-\Delta \mathrm{OD}}{0.07}\times \frac{V\times V^{\prime }}{v\times m}$$
where ΔOD represents the average change value of OD in the sample tube per minute for 4 min. *V* represents the total volume of sample extract. *V′* represents the total volume of reaction solution. *V* represents the volume of the sample extract contained in the reaction solution. *M* represents the fresh weight of the sample used in the sample extract.

Catalase (CAT) activity was determined by decomposition of H_2_O_2_ and was measured spectrophotometrically by assessing the decrease in absorbance at 240 nm [40]. In brief, 3 mL of phosphate buffers, 0.3 mL of enzyme solution, 2 mL of distilled water and 0.6 mL of 0.1 M H_2_O_2_ were added to the test tube and immediately shaken. Then, the ΔOD was quickly determined using the colorimetric method at 260 nm. The enzyme solution was replaced with phosphate buffer as the control. ΔOD was determined spectrophotometrically at 325 nm for 4 min and recorded every 30 s. The CAT activity was calculated using the following formula:$$\mathrm{CAT}\;\mathrm{activity}\;=\frac{\Delta \mathrm{OD}\times V}{0.01\times v\times m}$$
where ΔOD represents the average change value of OD in the sample tube per minute over 4 min. *V* represents the total volume of sample extract. *v* represents the volume of the sample extract contained in the reaction solution. *m* represents the fresh weight of the sample used in the sample extract.

Peroxidase (POD) activity was determined by the degree of oxidation of guaiacol by the spectrophotometer at 470 nm [39]. In brief, 2 mL of phosphate buffers, 1 mL of enzyme solution, 1 mL of 0.05 M guaiacol, and 1 mL of 2% H_2_O_2_ were added to the test tube and immediately shaken. ΔOD was determined spectrophotometrically at 470 nm for 4 min and recorded every 30 s. The POD activity was calculated using the following formula:$$\mathrm{POD}\;\mathrm{activity}\;=\frac{\Delta \mathrm{OD}\times V}{0.01\times v\times m}$$
where ΔOD represents the average change value of OD in the sample tube per minute over 4 min. *V* represents the total volume of sample extract. *v* represents the volume of the sample extract contained in the reaction solution. *m* represents the fresh weight of the sample used in the sample extract.

All the indices of samples, as well as the control experiments were tested with independent replicates.

### 4.5. Identification of ALMT Family Genes in C. sinensis

The amino acid sequence of *C. sinensis* genomic coding sequences (http://tpia.teaplant.org/index.html) (accessed on 23 January 2021) was compared with the reported *ALMT* gene family of *A. thaliana* (https://www.arabidopsis.org/) (accessed on 23 January 2021) by using Local BLASTp and Bioedit (v8.1.0, Manchester, UK). Furthermore, Pfam protein analysis online services (http://pfam.xfam.org/) (accessed on 23 January 2021) and SMART online services (http://smart.embl-heidelberg.de/) (accessed on 23 January 2021) were used to verify the candidate ALMT genes of *C. sinensis*, and finally the members of *CsALMTs* family were identified.

### 4.6. Bioinformatics Analysis for CsALMTs

ClustalW was used to compare the domains of *CsALMTs*, *AtALMTs* and *OsALMTs*, and MEGA 7.0 was used to construct the phylogenetic tree by using the neighbor-joining method and related parameters (Poisson model, pairwise deletion and 1000 bootstrap replications) [41]. According to the evolutionary relationship and referring to the classification of *AtALMTs* sequences, all identified *CsALMTs* were classified and named. The standardized naming rules were adopted in the present study. Briefly, each of the ALMT gene sequences was named using a prefix, ‘Cs’ (*Camellia sinensis*), and distinguished by additional numbers, such as *CsALMT1*, *CsALMT2* and soon on.

The basic physical and chemical properties, such as amino acid number and average hydrophobicity of *CsALMTs* protein sequence in *C. sinensis*, were predicted by using the ExPASy-ProtParam online services (https://web.expasy.org/protparam/) (accessed on 23 January 2021). Meanwhile, SoftBerry ProtComp 9.0 (http://linux1.softberry.com/berry.phtml) (accessed on 23 January 2021) and TMHMM Server.2.0 (http://www.cbs.dtu.dk/services/TMHMM) (accessed on 23 January 2021) were used to complete subcellular localization of *CsALMT* proteins and predict their transmembrane domains. SOPMA online services (https://npsa-prabi.ibcp.fr/cgibin/npsa_automat.pl?Page=npsa_sopma.html) (accessed on 23 January 2021) were used to predict the secondary structure of *CsALMT* proteins.

According to the conserved structure of *CsALMTs* and the annotation file of the *C. sinensis* genome (http://www.plantkingdomgdb.com/tea_tree/data/gff3/) (accessed on 23 January 2021), the schematic diagram of *CsALMTs* gene structure was constructed by using The TBtools online service (https://github.com/CJ-Chen/TBtools) (accessed on 23 January 2021). The MEME online service (http://meme-suite.org/tools/meme) (accessed on 23 January 2021) was used to identify the conserved motif (E-Value < 20) of the *CsALMT* amino acid sequence, and TBtools was used to construct the structural map of CsALMTs.

### 4.7. Total RNA Extraction and RT-qPCR Analysis

Total RNA was extracted from pollen tubes as described in item 4.1 of three independent biological replicates for each treatment and different *C. sinensis* tissues (root, stem, leaf and flower) of 5-year-old *C. sinensis* cv. ‘Longjing43’ without treatment using EASYspin Plus Complex Plant RNA Kit (Aidlab, Beijing, China, Cat No. RN53) following the manufacturer’s protocol. The quality and integrity of total RNA was checked with reference to the method of Wang et al. [8]. For RT-qPCR analysis of *CsALMTs*, HiScript^®^ III RT SuperMix for qPCR with gDNA wiper (Vazyme, Nanjing, China, Cat No. R323-01) was used to synthesize the first-strand cDNA. RT-qPCR analysis was conducted using the ChanQ^®^ SYBR qPCR Master Mix (Vazyme, Nanjing, China, Cat No. Q311-02) with the specific primer pairs shown in Appendix A. *β-Actin* served as a reference gene [42].

All the RT-qPCR tests were performed on the Bio-Rad BFX96 fluorescence (Bio-Rad C1000 TouchTM Thermal Cycler, Hercules, CA, USA). Each sample was run in three technical triplicates with three biological replicates. The specificity was confirmed by the melting-curve analysis of the amplified products at the end of the RT-qPCR test. The expression levels of *CsALMTs* were normalized to the *β-actin* based on the 2^-∆∆Ct^ method [43].

### 4.8. Statistical Analysis

Data analysis and correlation analysis were performed using SPSS software (SPSS Inc. version 22.0, Chicago, IL, USA, 2013) with Duncan’s test. The data diagrams were drawn with SigmaPlot software (SigmaPlot, version 12.5, Systat Software Inc., San Jose, CA, USA) and R software (R, version 4.1.0, Auckland, New Zealand, 2021).

## Figures and Tables

**Figure 1 plants-11-02233-f001:**
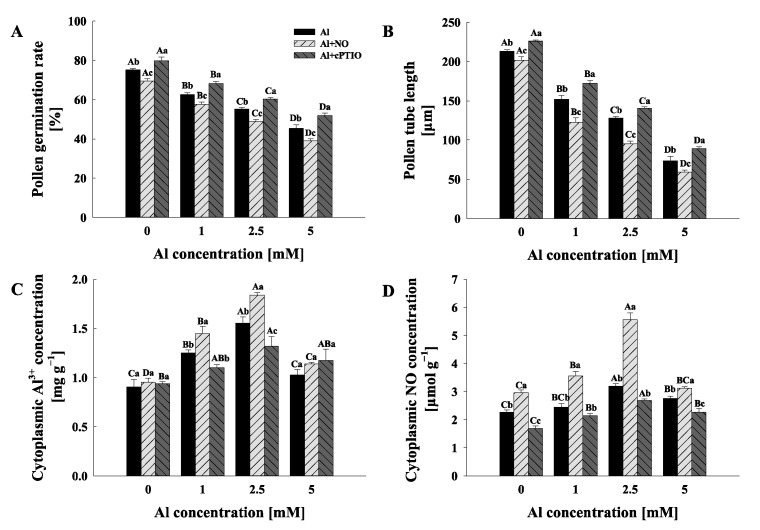
The effects of different treatments on the pollen germination rate (**A**), pollen tube length, (**B**) cytoplasmic Al^3+^ concentration, (**C**) and cytoplasmic NO concentration (**D**) of *Camellia sinensis*. Al represents the treatment group treated with Al concentration gradient only, Al + NO represents the treatment group treated with Al^3+^ concentration gradient treatment and 25 μM DEA NONOate (NO donor), and Al + cPTIO represents the treatment group treated with Al^3+^ concentration gradient treatment and 200 μM carboxy PTIO potassium salt (NO scavenger). Different lowercase letters represent significant differences among different nitric oxide conditions under the same Al^3+^ concentration treatment, and different uppercase letters represent significant differences among different Al^3+^ concentration treatments under the same nitric oxide conditions (for pollen germination rate and pollen tube length, n = 10, and for cytoplasmic Al^3+^ and NO, n = 3, *p* < 0.05), as determined by the Duncan test.

**Figure 2 plants-11-02233-f002:**
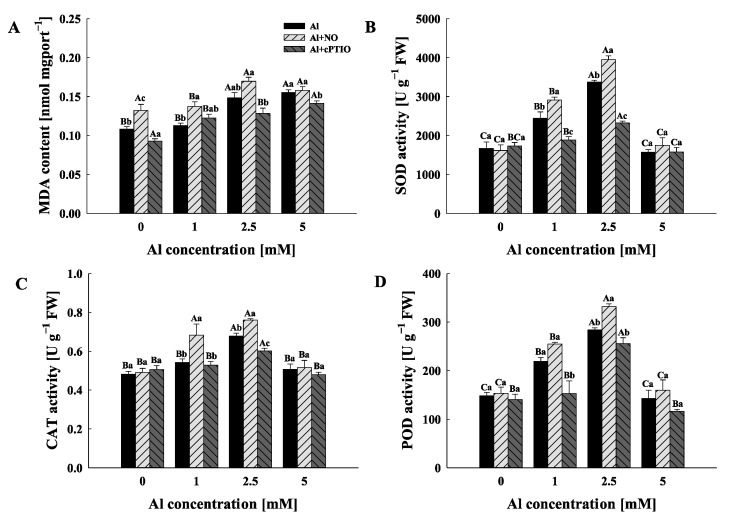
The effects of different treatments on the malondialdehyde (MDA) contents (**A**), superoxide dismutase (SOD) (**B**), catalase (CAT) (**C**) and peroxidase (POD) (**D**) activities of *Camellia sinensis* pollen tubes. FW: fresh weight. Al represents the treatment group treated with Al concentration gradient only, Al + NO represents the treatment group treated with Al^3+^ concentration gradient treatment and 25 μM DEA NONOate (NO donor), and Al + cPTIO represents the treatment group treated with Al^3+^ concentration gradient treatment and 200 μM carboxy PTIO potassium salt (NO scavenger). Different lowercase letters represent significant differences among different nitric oxide conditions under the same Al^3+^ concentration treatment, and different uppercase letters represent significant differences among different Al^3+^ concentration treatments under the same nitric oxide condition (n = 3, *p* < 0.05), as determined by the Duncan test.

**Figure 3 plants-11-02233-f003:**
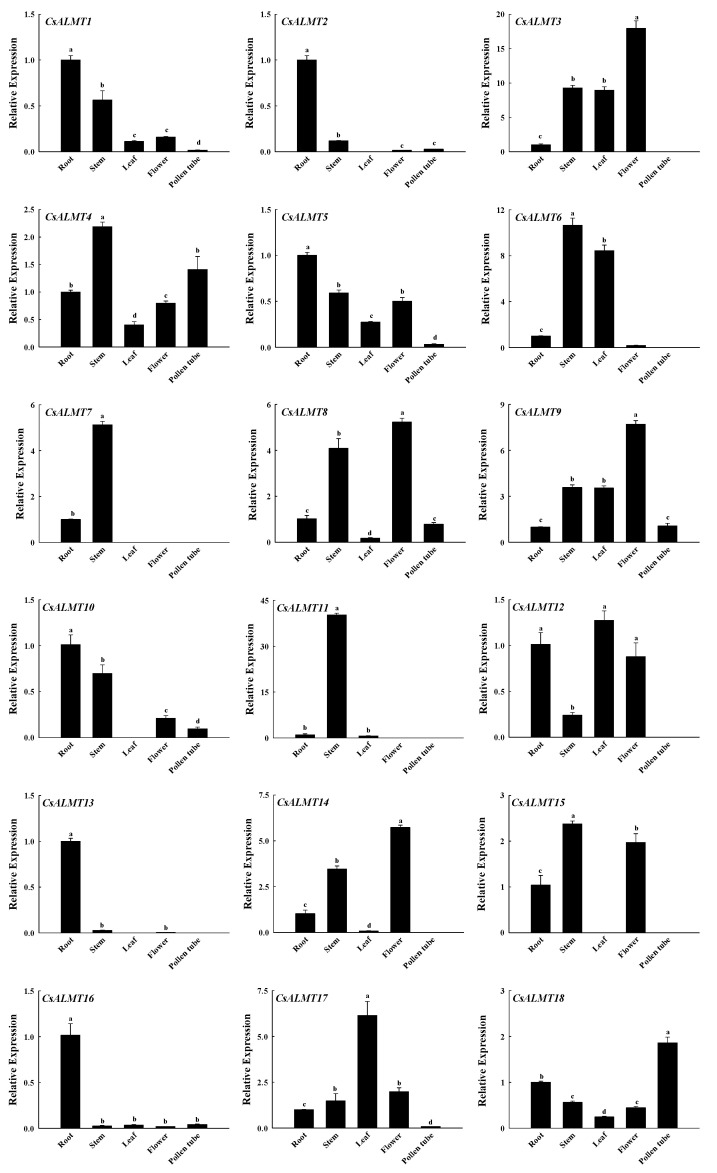
The expression level of *CsALMTs* in different tissues of *Camellia sinensis*. Different lowercase letters represent significant differences among different tissues (n = 3, *p* < 0.05), as determined by the Duncan test.

**Figure 4 plants-11-02233-f004:**
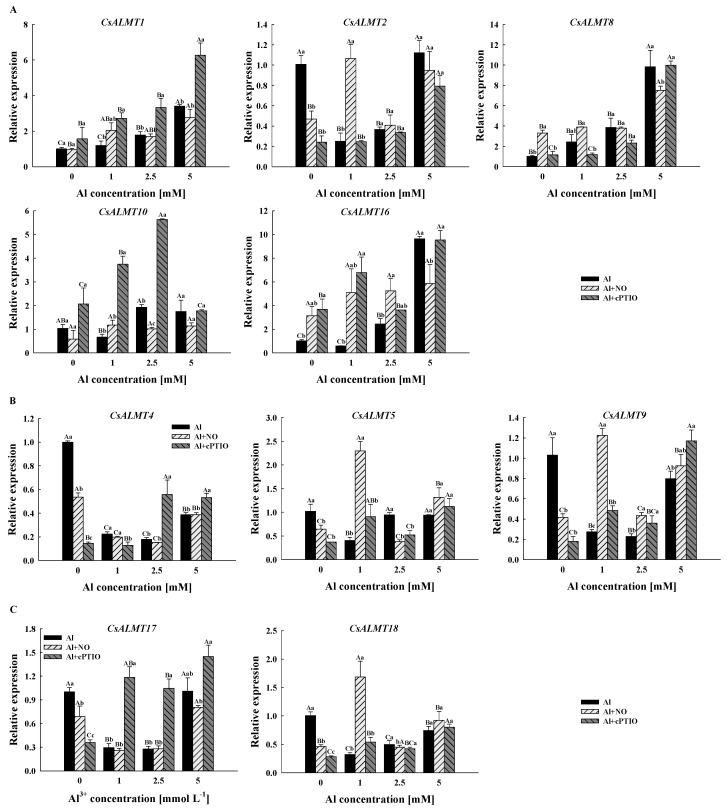
The relative expression levels of *CsALMTs* belong to subclass I (**A**), subclass II (**B**) and subclass IV (**C**) in *Camellia sinensis* pollen tubes under different treatments. Al concentration represents the Al^3+^ concentration gradient in the culture medium, Al treatment represents the treatment group treated with Al concentration gradient only, Al + NO represents the treatment group treated with Al^3+^ concentration gradient treatment and 25 μM DEA NONOate (nitric oxide donor), Al + cPTIO represents the treatment group treated with Al^3+^ concentration gradient treatment and 200 μM carboxy PTIO potassium salt (nitric oxide scavenger). Different lowercase letters represent significant differences among different nitric oxide conditions under the same Al^3+^ concentration treatment, and different uppercase letters represent significant differences among different Al^3+^ concentration treatments under the same nitric oxide conditions (n = 3, *p* < 0.05), as determined by the Duncan test.

**Figure 5 plants-11-02233-f005:**
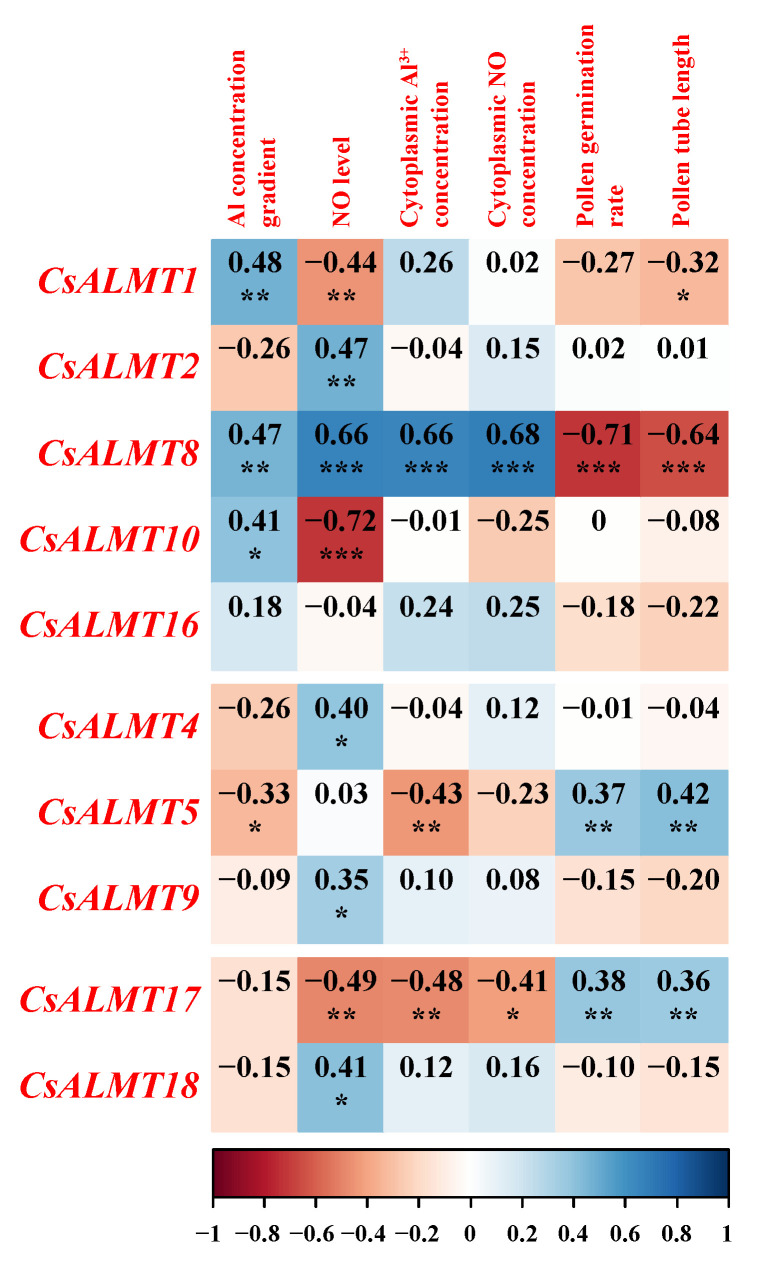
The modified (the data under 5 mM Al^3+^ treatment were removed) Pearson correlation coefficients between Al^3+^ concentration gradient and NO level in the culture medium, cytoplasmic Al^3+^ and NO concentration, pollen germination rates, pollen tube lengths and *CsALMTs* expression levels. Significant differences are reported as * (0.01 < *p* < 0.05), ** (0.001 < *p* < 0.01) and *** (*p* < 0.001), as determined by the Duncan test.

**Figure 6 plants-11-02233-f006:**
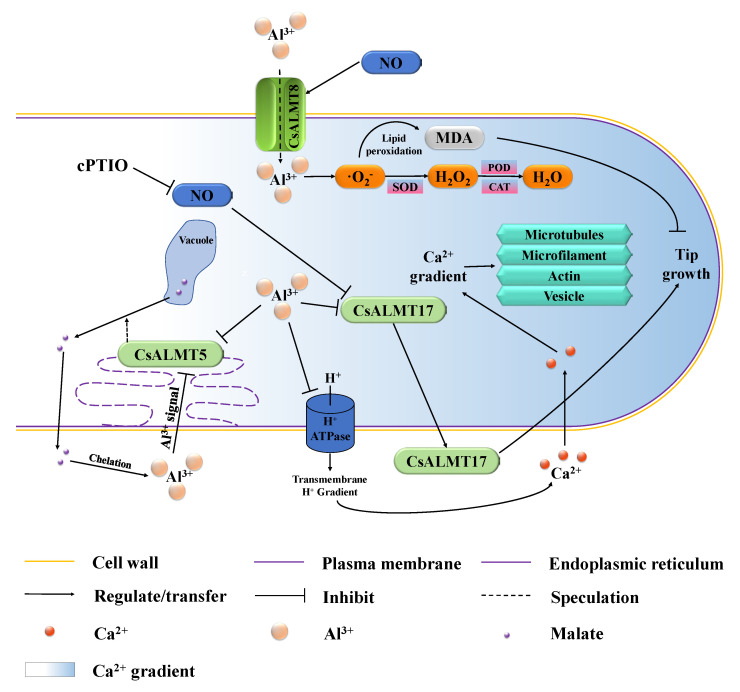
Potential hypothesis model that nitric oxide (NO) participates in exogenous Al^3+^ inhibition of *C. sinensis* pollen tube growth by regulating *CsALMTs*.

**Table 1 plants-11-02233-t001:** Two-way ANOVA test *F* value of pollen germination rates (PGRs), pollen tube lengths (PTLs), cytoplasmic Al^3+^ (C_Al_) and NO (C_NO_)concentrations, malondialdehyde (MDA) contents, antioxidant enzyme activities and relative expression level of *CsALMTs* for *C. sinensis* cv ‘Longjing43’ affected by Al concentration gradient and NO level in the culture medium.

Indicator	Al Concentration Gradient	NO Level	Al Concentration Gradient × NO Level
PGR	360.73 ***	98.54 ***	0.33 ns
PTL	808.36 ***	110.99 ***	3.57 *
C_Al_	59.20 ***	12.95 ***	5.01 **
C_NO_	93.57 ***	202.58 ***	19.51 ***
MDA	45.33 ***	31.79 ***	3.25 *
SOD	120.27 ***	32.78 ***	11.95 ***
CAT	35.32 ***	11.38 ***	3.41 *
POD	85.13 ***	19.97 ***	2.43 ns
*CsALMT1*	31.707 ***	23.293 ***	3.477 *
*CsALMT2*	19.830 ***	13.046 ***	10.070 ***
*CsALMT4*	32.305 ***	7.324 **	23.334 ***
*CsALMT5*	50.007 ***	11.831 ***	46.449 ***
*CsALMT8*	86.568 ***	2.478 ns	4.492 **
*CsALMT9*	14.940 ***	11.086 ***	18.032 ***
*CsALMT10*	16.988 ***	71.364 ***	11.985 ***
*CsALMT16*	19.636 ***	6.496 **	4.385 **
*CsALMT17*	20.919 ***	29.545 ***	15.241 ***
*CsALMT18*	9.824 ***	12.570 ***	16.588 ***

Significant differences are reported as ns, *p* > 0.05. * *p* < 0.05. ** *p* < 0.01. *** *p* < 0.001.

## Data Availability

All data generated or analyzed during this study are included in this published article (and its Appendix A).

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
