# Peer review of "Nitric Oxide Participates in Aluminum-Stress-Induced Pollen Tube Growth Inhibition in Tea (Camelliasinensis) by Regulating CsALMTs"

_plants, 2022, doi:10.3390/plants11172233_

Round 1

Reviewer 1 Report (Previous Reviewer 1)

The manuscript was improved significantly, clarifying the treatments and part of the results. However, a deep English review are still required in the whole manuscript.

Several questions and suggestions were included along the manuscript, which were complemented in the comments below.

Introduction:

The paragraph starting with “Exploring …” aim at introducing the ALMT in the context of the manuscript. First of all, this paragraph should be reworded in order to clearly justify the importance of ALMT to your manuscript. The first reference (18) is a review describing the role of Mg in the Al toxicity. In my opinion it is not adequate to describe the achievements of ALMT as an Al activated organic acid transporter related to Al tolerance.

In your answer, it was mentioned your previous results on CsALMT, which should be added in you introduction to clarify the scope of the current work.

Results:

The item 2.4 describes several bioinformatics strategies to characterize the ALMT members, but the most important point is to select what genes/proteins were involved in Al tolerance mechanisms in Arabidopsis and rice. This information will be very useful for any further result and discussion. This issue would also help on focusing your discussion to the target genes.

The expression of the CsALMTs (item 2.5) should be improved and corrected. As the pollen tissue was included in the fig 3, the text should be revised to include all tissues in the context. The high expression level of 1 should be explained, because in my opinion, 1 can not be considered ahigh expression level.

The item 2.6: I don´t have specific comments but in general the text is confused comparing the expression pattern of different pairs of CsALMTs, and starting most of the phares with conjunctions (however, furthermore, in addition, etc). In my opinion, phrases describing different genes do not need to be linked with conjunctions.

The item 2.7 has also to the reworded. Long phrase, genes with similar expression pattern that was not cited, a negative correlation that was describe as positive. Please revise and consider for further discussion.

Discussion:

After the reformulation of the manuscript, the results were clarified and several missing points appeared in the discussion. There were too many speculations about gene function along the discussion, based on each fragment of the result. These points were highlighted along the text. However, a final and consistent speculation would be expected upon a deep discussion of all results, at least for each target gene presented in your final hypothesis (Fig 6).

Author Response

Reviewer 2 Report (New Reviewer)

I have reviewed the MS entitled “Nitric Oxide Participates in Aluminum Stress-induced Pollen Tube Growth Inhibition in Tea (Camellia sinensis) by Regulating CsALMTs’’.  This paper has already been reviewed. 

I found that the topic of the paper is interesting, since Camellia sinensis is an Al accumulator plant, the fact that pollen germination and pollen tube development are sensitive to aluminum is an important physiological consideration.  

The main purpose of the paper is getting knowledge in the NO effect in the Al stress pollen tubes of tea plants and relate this effect with the expression of ALMT transporters.  They did experiments in pollen germination and pollen tube growth with aluminum, and with a donor and a scavenger of NO, to increase or decrease the NO concentration. They measured lipid peroxidation and antioxidant enzyme activity; and finally, they performed an extensive bioinformatic analysis to find the ALMTs involved in the response of the C. sinensis pollen to Al toxicity, and check they expression in roots, stems, leaves, flowers, and pollen tubes.

I consider that the authors made most of the suggestions and recommendations of the former reviewer.  However, I believe that there are still some points to be reviewed: 

Introduction section:  In page 2 it says: ‘’ Although the mechanism of Al toxicity in pollen tubes people have been deeply studied’’.  It is not clearly explained, or not very understandable.

I suggest that the background of the study needs to deep into the role of Calcium in pollen tube growth, they mentioned it in the discussion section (Fang, et al., 2019), but I consider it is important to include this information in the introduction.  Does Al interfere with calcium gradient? It is well known that Al can compete with Calcium, so it will be very important to examine this concept.

Another suggestion is that they need a hypothesis, perhaps explaining their model (Fig. 6).  My general suggestion is clearly to point out what is the main purpose to analyze the role of NO and the ALMT transporters.

Materials and Methods section:  I consider that they followed well the recommendations of the previous reviewed, since the methods are well described. 

My suggestion is to give more information on the enzyme activities:  number of replicates of the measurements, how they determine the activity (formulas), and so on.

In qPCR analysis, why do they use only one reference gene?  To support more the expression levels, generally we need to use at least 2 genes, b-actin is a well-known reference gene but sometimes during treatments its expression may change. 

Results: I found them well described, I just have minor corrections: when the authors mention or call to figures, they need to include specifically if it is Fig. 1 A, B, …, not only to mention the number of the figure (see page 3). Figure legend of Fig. 4, needs to explain the meaning of upper- and lower-case letter (statistically), and also need to give details in the number of replicates, bars are standard deviation or error?

Discussion:  my general suggestion is not to be too speculative: page 12, ‘’Since CsALMT17 is subcellularly localized outside the cell, we speculate that it may be a secreted protein that starts from the cytoplasmic matrix and is secreted or transported to the outside of the cell after being synthesized by ribosomes on the surface of the endoplasmic reticulum’’.  The results seem robust, but they need to do more experiments to confirm the role of NO in the expression of these ALMT genes. What about the promoters of these genes?  How they will confirm that ALMT 5, 9, 17 and 18 are participating in the Al tolerance? and are involved in the signaling pathway of NO?  How will they clarify if NO is a negative regulator?

Author Response

Reviewer 3 Report (New Reviewer)

The revised manuscript by Xu et al seems to be extensively modified. Thus it is thought to be accepted for publication in Plants except for minor changes as follows:

1) P 2 L12: changed as follows: ... studied. Up to now, the mitigation... to studied up to now, the mitigation...

2) P2 L22 to 24: where two successive 'One the other hand' 

3)P2 L32: ALMT:aluminum activated malate "transporter" but not "protein"

3) P2 L32 and so on: Generally protein name shows as upright and gene and botanical names as Italic. There are too many mistakes for them to count all, for examples, ALMT, should be upright at P2 L32 and ALMT, ALS1 and ALS1 and ALS3, Nrat1, all should be upright. In every page the same kind of mistakes can be found. The Authors survey all the sentences in the manuscript on this matter. Thus they all should be correctly expressed.

4)P3 L16: change higher to lower.

5)P7: 'Fig. 4' is missing to insert somewhere in second or third paragraph.

6) P9 L5: no space 'with' and 'out'.

7) P11 2nd and 3rd paragraphs: References 1 and 2 are not correct.

8) Figure 6: CsALTM5, 8, 15 and 17 should be upright and 'ATPase' and 'growth'.

9) Figure S2 and S4: CsALMTs (not Italic)

10) Table S1: CsALMT17 has five transmembrane domains? 'Transmembrane' but not 'Transmembra ne'.

Round 2

Reviewer 1 Report (Previous Reviewer 1)

A few editing are still required, such as Al 3+ superscript. I don't additional skills for english review

This manuscript is a resubmission of an earlier submission. The following is a list of the peer review reports and author responses from that submission.

Round 1

Reviewer 1 Report

Methods: The description of the treatments is very confused. Avoid all acronyms such as GAl, LNO, AT, NT, CT and GT. These acronyms should be eliminated of the entire manuscript and replaced by a clear description of the treatments. The major point of the manuscript is to evaluate the combination of Al x NO concentrations.

For example: normal NO = Al treatment only or without NO, high NO = Al+NO and low NO = Al + cPTIO, but explaining why cPTIO is low NO. Finally, simplify and clarify your experimental design in the Methods and present all components of each treatment in the whole manuscript, including figures and results, instead the acronyms.

Introduction: Lines 72 – 84: The introduction of Al tolerance genes should be better organized. The transcription factors are mixed with organic acid transporter genes associated with Al tolerance mechanisms. The authors have to decide what is the target point that should be introduced to understand the selection of the ALMT gene family to be evaluated in the manuscript. This point is not clear. Several Al tolerance genes are conserved among different plant species, such as MATE, WRKY, STOP/ART, Nrat1… This study was carried out in pollen, whereas all ALMT genes responsible for Al tolerance in plants are induced by Al in root tips, transporting malate to the rhizosphere. The involvement of ALMT should evaluate the malate transport activity. Thus, my major question remains: is why ALMT family was selected for this study?

Lines 92 – 100: This paragraph seems to be comments and suggestions for the introduction that were left. Please delete and, if pertinent, include the comments in the text.

Results: The text was hard to read with all acronyms instead of the description of each treatment. I read the manuscript with a list of acronyms side by side. I made some suggestions, but the text should be carefully revised based on the new description of the experimental design, as suggested above.

In Bioinformatics topic, why to use all ALMT members of Arabidopsis and rice, instead of only the genes that were characterized as involved in Al tolerance in these species, or members associated with other traits? The use of members without functional characterization has no meaning in a phylogenetic tree.

Figure S2 requires several explanations and corrections: “Cluter” should be replaced to subclasses, which is cited along the text. The red arrows should be replaced by the bootstrap numbers because the arrow size cannot be differentiated in the figure. Reducing the number of At and Os members not functionally characterized would improve the figure presentation.

Line 194: the subitem 2.4.4 should not be under "Bioinformatics" once it is an experimental result of qPCR.

In figures 3 and 4, the color gradient should be replaced by numbers. Imagine a reader color blind or with black and white printer?

Figure 3 is a general description of expression pattern of the CsALMT genes in the plant. Thus, I would suggest to include the pollen tube, which is the target tissue of the research. Additionally, the next section (lines 209-211) describes that only 10 CsALMTs were expressed in pollen tubes, which were not shown. The best option is to include this tissue in Fig 3.

My interpretation on the negative expression values in Fig 3 is that several ALMTs were less expressed than the internal control gene. First of all, it is important to describe what was the internal control gene, which is not cited in the methods. Then, clearly explain what means these results. After changing the colors by numbers, the expression levels could be clearly presented. For example, in my point of view none of the ALMT genes were “highly” expressed in roots, this column is completely white or off colored. This is very different from what is written in lines 201-203 (genes highly expressed in root). I also disagree in other interpretations of these expression patterns, please revise this section.

Figure 4: A very important figure, but poorly presented. There is a multitude of colors and acronyms that brings a great confusion for a complex combination of Al x NO treatment. This figure should be completely reorganized and replaced by a table, with numbers and a clear description of the treatments. 1) Replace all acronyms by the description of treatments and concentrations; 2) Replace the internal colors by numbers; 3) A table would be more informative than a figure; 4) The NO effect cannot be extracted from this figure.

The term "Al concentration" is a much clear information than GAl, the same for LNO. For example, [Al3+] is cited along the text as Al concentration internal and external. Please, clearly describe all traits.

Item 2.5: When line 212 start with “AT, NT and CT”, no clear information can be extracted, once these acronyms describe all treatment combinations between Al x NO. It is not clear to me how these data could be presented, but I am sure that is not as shown in Fig 4.

The Figure 4 does not show the effect of NO, but mainly the Al.

Lines 213-214: The authors affirm that genes CsALMT 1 and 8 have different expression under NO and Al. How different? The fig 4 shows a similar expression trend for CsALMT8 expression in all three combinations of AlxNO and the fig 5 shows similar correlation coefficient for the CsALMT8 expression with Al and NO levels. A similar expression trend is also observed for CsALMT1 (Fig 4), whereas the correlation coefficients have opposite sign for this gene expression under Al and NO (Fig 5). Thus, what data we need to follow?

Another important point is the significance, which was cited in the text, but not presented. Probably, after reformulating fig 4, it would be possible to include the significance levels of each treatment combination.

Several notes were included in the text, and a deep reformulation of the item 2.5 is needed.

Discussion: Interestingly, the figure 4 is not discussed and figure 5 is firstly cited in the discussion. It is important to explain how the correlations were performed and to present the figure 5 in the results. As previously questioned in the item 2.5, the results from figs 4 and 5 should be consistent.

Phrase starting in line 311: The phrase has to be contextualized with your data, improve the discussion.

Line 313: Add another paragraph, as another subject.

Phrase starting in line 313: Long phrase, which should be reworded. The correlations without the data of 5 mM would bring a different result from figure 4, whereas both figs should reflect similar results.

Phrase starting in line 322 is also long, and should be reworded.

Lines 323: As previously comment, the effect of NO was not clearly shown in figure 4, which is very important for the overall discussion. Additionally, it is important to explain how the correlation coefficients were calculated, mainly for the different levels of NO (LNO).

Line 324 - 325: It is interesting to point out that, based on the Fig 5, the expression of CsALMT8 was the only gene positively correlated Al, NO, [Al] and [NO], and negatively correlated with the pollen tube elongation indices. However, to declare CsALMT8 “as a negative regulator of the response of C. sinensis pollen tube to Al stress” is not adequate, once correlations coefficients cannot infer causal relationship.

Lines 327-329: It is important to highlight that differential expressions or correlation coefficients cannot support functional evidences for the CsALMT candidate genes. However, the positive correlation between CsALMT8 expression and Al concentration is the same expression pattern of all ALMT genes validated as Al tolerance. In fact, these ALMTs were activated in root tips in the presence of Al as a malate transporter. Thus, the functional assign, the comparison, and the term “opposite function” are incorrect and should be revised by the authors

Line 332: An opposite description compared with the figure 5.

As several observations were made along the manuscript, the paragraph starting in the line 345 has to be revised, as well as the Figure 6.
